# Transcriptome and Biochemical Analysis of the Mechanism of Low-Temperature Germination in *Acer truncatum* Bunge Seeds

**DOI:** 10.3390/ijms262211193

**Published:** 2025-11-19

**Authors:** Huijing Meng, Linpo Zhou, Yiming Qin, Shuang Ji, Pengpeng Wang, Yufan Liu, Jiawen Liu, Jingyu Ma, Hexiang Sun, Xiuhong Zhu, Guangxin Ru

**Affiliations:** College of Forestry, Henan Agricultural University, Zhengzhou 450046, China; 15938277261@163.com (H.M.);

**Keywords:** *A. truncatum* Bunge, transcriptome, low temperature, biochemical, germination

## Abstract

*Acer truncatum* Bunge exhibits remarkable cold tolerance at the mature seedling stage, yet the mechanisms governing its seed germination under low-temperature conditions remain poorly understood. To elucidate the molecular and physiological mechanisms underlying low-temperature germination in *A. truncatum* seeds, we selected *A. truncatum* seeds as the experimental material. The seeds were evenly divided into two groups and subjected to germination under 25 °C (control) and 4 °C (low-temperature stress) conditions, followed by transcriptome sequencing and physiological and biochemical analyses. Transcriptome sequencing analyzed differential genes and physiological indicators. Fourteen transcription factor families were identified (ARR-B, AP2-EREBP, bHLH, NAC, FAR1, MADS, WRKY, AB13VP1, bZIP, C3H, CROS, LOB, TCP, and SBP). These regulate seed germination under abiotic/biotic stress. GO term enrichment occurred in biological processes. KEGG enrichment involved carbon metabolism, the glutathione pathway, the citrate cycle, and glycolysis. Most genes were upregulated. Citrate cycle and glycolysis correlated with seed activity, promoting germination. The glutathione cycle greatly improves the stress resistance of seed germination. There were 1804 genes that were upregulated and 8075 genes that were downregulated during seed germination. Among differential genes, CBF 5 was significantly downregulated but most WRKY families and LEA14-A were upregulated to maintain cell homeostasis. Meanwhile, GSH, SOD, POD, and proline (Pro) levels increased with prolonged stress. MDA rose initially, then declined. Soluble protein content first increased, then decreased, but remained higher than controls. Seeds germinated under low temperature, but germination potential was slightly lower than at room temperature. We propose that LEA protein, antioxidant enzymes, and Pro accumulation enhance cold tolerance. This study elucidates the physiological and molecular mechanisms underlying seed germination, advancing the understanding of cold tolerance in *A. truncatum*.

## 1. Introduction

For most temperate and subarctic plant species, low-temperature exposure constitutes the predominant environmental stress factor during winter dormancy [1]. *A. truncatum* was cultivated in Chifeng City, Inner Mongolia, characterized by winter temperatures averaging 0 °C. Despite these environmental conditions, *A. truncatum.* demonstrated typical germination and growth patterns [2]. Previous investigations have elucidated the genomic underpinnings of cold tolerance in *A. truncatum*, identifying WRKY family genes as pivotal transcription factors [3,4]. However, the mechanisms regulating low-temperature germination in *A. truncatum* seeds remain uncharacterized, representing a significant lacuna in the comprehension of cold stress responses. Therefore, *A. truncatum* seeds were procured from Chifeng City, Inner Mongolia, for this research to fill the gap in the response mechanism of *A. truncatum* seeds to low temperatures during this stage of its life cycle.

Seed germination represents a sophisticated physiological process initiated by water imbibition and seed swelling, followed by embryonic development and testa rupture, culminating in radicle emergence which signifies successful germination. This biological process encompasses a series of coordinated physiological and biochemical alterations, including differential gene expression, de novo protein synthesis, and extensive post-translational modifications [5]. Seed germination is regulated by endogenous factors such as phytohormones and endosperm decay and external environmental factors such as light, temperature, water, and oxygen. Temperature is an extremely important factor to determine seed germination [6]. The suitable germination temperature of most crops is 15~30 °C; temperatures beyond the suitable temperature range are not conducive to water absorption and physiological metabolism, resulting in seed germination being blocked [7].

As the primary stage of morphological formation, seed germination is first exposed to low temperature and the most susceptible to low temperature. Low temperature slows down the water absorption rate of seeds, resulting in delayed germination, and aggravates the adverse effects of other biotic and abiotic factors in the soil on germination, resulting in the decline in germination rate and emergence rate, affecting crop growth and development in the later stage, and ultimately reducing crop yield [8,9]. Low temperature hinders rapid seed germination, which is not conducive to simplified direct cultivation of crops. Rice (*Oryzasativa*) and maize (*Zea mays*) are prone to low temperature stress in early spring during spring sowing, and encounter low-temperature ribs in autumn and winter, inhibiting seed germination. Especially due to the intensification of rice-oil progress, late live broadcast rape is more prone to low-temperature invasion and seed germination failure, resulting in large areas of cultivated land remaining idle in winter [10]. In addition, the low-temperature inhibition of seed germination limits the geographical distribution of crops, which makes convenience crops difficult to grow in alpine zones. Therefore, for the purpose of stable agricultural production, it is important to explore the physiological characteristics and molecular regulation mechanism of low-temperature tolerance for breeding varieties with excellent low-temperature tolerance, making full use of idle winter fields and broadening the crop planting area.

Previous studies have shown that freezing stress can trigger the activation of defense systems in plants. The peroxidase (POD), catalase (CAT), superoxide dismutase (SOD), and (GSH) glutathione protective enzyme activities are enhanced, reducing the accumulation of reactive oxygen species (ROS) and reducing the freezing stress damage to plants [11]. Furthermore, the accumulation of certain soluble compounds can mitigate both biotic and abiotic stresses; for instance, proline (Pro). Virtually all biological and abiotic stressors (including, but not limited to, low temperature, high temperature, high salinity, drought, and pathogen/pest infestations) induce an increase in proline content within the plant [12]. This increase plays a crucial role in stabilizing cellular protein structure, reducing cellular acidity, and scavenging reactive oxygen species, thereby diminishing stress-induced damage and maintaining normal plant growth and development [13]. Moreover, the accumulation levels of Pro exhibit variability across different plant species. Following the alleviation of stress, proline undergoes rapid degradation and utilization, participating in the synthesis of chlorophyll and other essential compounds. This process facilitates the rapid recovery of plant growth and reduces the accumulation of reactive oxygen species and malondialdehyde (MDA), among other substances [14].

Low-temperature germination (LTG) is a complex quantitative trait controlled by multiple genes with extensive variation in the crop germplasm [15,16]. Although the mechanism of plant cold response has been widely studied, the mechanism of cold response at seed germination is not clear. Moreover, genetic studies of low-temperature germination progress slowly compared with other agronomic traits. To this day, gene loci for cold seed germination for quantitative traits (quantitative trait loci, QTLs) were only initially detected in rice, maize, soybean (Glycine max), rape, sorghum (Sorghum bicolor), cucumber (Cucumis sativus), and lettuce (Lactuca sativa) crops. Analyses of gene cloning and expression regulation are scarce [17,18,19,20,21,22]. Therefore, it is urgent to identify the key genes of seed low-temperature germination tolerance, understand the genetic regulation mechanism, and realize the rapid germination of seeds under low-temperature conditions through genetic improvement. In this study, the changes in transcriptome and physiological indicators of *A. truncatum* seeds at low temperature were analyzed, in order to provide a reference for the cloning of low-temperature tolerance germination genes, molecular mechanism analysis, and the improvement of low-temperature tolerance traits of crops.

## 2. Results

### 2.1. Physiological Changes of A. truncatum Seeds Under Low-Temperature Stress

In this experiment, seeds were assessed under both low-temperature and normal conditions, revealing a marginally reduced germination potential under low-temperature conditions compared to normal temperature (Figure 1A,B). The relative water content (RWC) of seeds under low-temperature conditions was slightly lower than that of seeds under room temperature, but no significant difference was observed between the two groups (Figure 1C). However, the soluble protein content and proline content in *A. truncatum* seeds germinated under low temperature were significantly higher than those in the control group. Plant protein content is closely correlated with relative water content (RWC).

### 2.2. Temporal Patterns of Soluble Osmolytes in Germinating A. truncatum Seeds

Plant cells typically enhance their water retention capacity by accumulating intracellular soluble compounds, thereby mitigating cellular damage under stress conditions. Pro is an important osmotic substance in the plant body, with the characteristics of water solubility, high water potential and small molecular weight [23]. Soluble proteins participate in plant metabolism. Their content changes significantly under stress conditions. Short-term low-temperature stress induces a temporary increase in soluble protein content [24]. Siddig [25] found that plant protoplasm a dehydrated under cold stress triggered the accumulation of Pro in the body to reduce the freezing point of plants and reduce the damage caused by low temperature to plant cells. With extended treatment duration, Pro content initially stabilized, followed by a gradual increase over 24 h, and ultimately peaked at 48 h, which significantly differed from other time points. There was no difference between 48 h and 72 h. The soluble protein content initially increased under low-temperature stress. It peaked at 24 h. After that, it gradually decreased with prolonged stress (Figure 2A,B).

### 2.3. Antioxidant Enzymes Play a Crucial Role in Aiding Plants to Eliminate Superoxide Free Radicals

Antioxidant enzymes play a crucial role in alleviating excess ROS. Reactive oxygen species (ROS) serve as crucial signaling molecules that enable rapid cellular responses to diverse stimuli. Appropriate ROS levels regulate plant responses to multiple stresses through signal transduction pathways, whereas excessive ROS accumulation induces oxidative stress. To investigate the mechanism of lower levels of H_2_O_2_ and O^2−^ in *A. truncatum* seeds, the activities of GST, superoxide dismutase (SOD), and peroxidase (POD) in seeds were measured under both normal and stress conditions. The results indicated that during short-term exposure to low temperatures, the activity of POD surged rapidly. After 48 h of such cold treatment, POD activity peaked (Figure 3A). The enzymatic activity changes of SOD leveled off after 48 h, followed by a subsequent decline. This suggests that while short-term low-temperature stress initially triggers an increase in plant SOD activity to mitigate damage, prolonged exposure to low temperatures results in a sustained decrease in its activity (Figure 3B).

The GST content was higher at low temperature than in the ingot maple seeds under control conditions (Figure 3C). Membrane lipid peroxidation, mediated by intermediate radicals, and the resultant malondialdehyde (MDA) accumulation, may compromise cytoprotective enzyme functionality and deplete antioxidant reserves, thus accelerating the propagation of membrane lipid peroxidation [26]. In this study, MDA content increased and then decreased. The content of MDA reached the highest at 12 h and then gradually decreased (Figure 3D). This indicates that there is an oxidative reaction in seeds at the beginning of low-temperature treatment leading to membrane lipid peroxidation in cells. With the prolongation of time, the self-regulation mechanism in the plant body alleviated the oxidative damage of plant cells.

### 2.4. Transcriptome Sequencing Results and Transcription Factor Analysis

Two groups underwent transcriptome sequencing, with each sample exhibiting a GC content greater than 43%, a Q20% exceeding 97%, and a Q30% surpassing 95% (Appendix A). These metrics met the criteria for proceeding with further biological analysis. We performed principal component analysis on two groups. This showed significant differences between groups and more reproducible samples within groups. At the same time, the differential genes between groups were compared, and Figure 3B shows how germination under low-temperature stress and normal temperature involves a large number of differential genes; the further the genes are from the middle of the figure, the more significant they are. Transcriptome sequencing revealed the presence of 56,732 genes of 832 transcription factors, which are categorized into 42 TF families (Figure 4C). The top five families were ARR-B (242), AP2-EREBP(113), BHLH (111), NAC(92), and FAR1 (89), among which ARR-B has the largest number of genes, and ARR-B genes may play an important regulatory role under low-temperature stress.

### 2.5. GO and KEGG Enrichment Analysis of the Differentially Expressed Genes (DEGs)

To gain insights into metabolic and gene function during seed germination, KEGG pathway and GO functional enrichment analyses were conducted. A total of 2583 unique genes were assigned to the top 20 of KEGG B class pathways, which were significantly enriched and encompassed “Carbon metabolism”, “Citrate cycle (TCA cycle)”, “Ribosome”, “Pyruvate metabolism”, “Alanine, aspartate and glutamate metabolism”, “Biosynthesis of amino acids”, “Glutathione metabolism”, “Purine metabolism”, “Glycolysis/Gluconeogenesis”, “Protein processing in endoplasmic reticulum”, and “Metabolic pathways” (Figure 5A). Among these, “Metabolism” featured the largest number of genes (1755), primarily in pathways such as “Carbohydrate metabolism”, “Biosynthesis of amino acids”, and “Protein processing in endoplasmic reticulum”. “Gluconeogenesis processing” followed, with 89 genes. A total of 74,891 unique genes were annotated in the GO database, revealing enrichment across 64 functional groups categorized under “Biological process”, “Cellular component”, and “Molecular function” (Figure 5B). Within the biological process category, twenty-six functional categories were enriched, with the most prominent ones including “Cellular process”, “Metabolic process”, “Response to stimulus”, “Regulation of biological process”, “Biological regulation”, “Cellular component organization or biogenesis”, “Localization”, “Single-organism process”, “Signaling”, “Establishment of localization”, “Developmental process”, and “Multicellular organismal process”. In the cellular components category, twenty-three functional groups demonstrated enrichment, including prominent ones such as “Cell”, “Cell part”, “Organelle, organelle part”, “Membrane, membrane part”, and “Macromolecular complex”. The molecular function category displayed enrichment in 15 functional groups, with “Binding and catalytic activity” being the most prevalent. Overall, the number of genes involved in biological processes was the highest, while the number of genes involved in molecular function was the lowest. 

### 2.6. Interplay Between the TCA Cycle and Glutathione Redox System in Stress Responses

Under low-temperature stress, seed gene expression primarily centered on the TCA cycle and glutathione cycle (Figure 6A). The TCA cycle further participated in the glutathione cycle, unsaturated fatty acid metabolism, and glycolytic pathways. This high metabolic engagement significantly enhanced seed vigor and promoted germination. In addition, the TCA cycle-derived α-ketoglutarate serves as a precursor for glutathione synthesis, while its NADPH output supports fatty acid desaturation (Figure 6B). This dual role synergistically enhances membrane stability (via unsaturated lipids) and oxidative stress tolerance (via GSH), collectively driving germination under cold stress. “ROS scavenging via the glutathione cycle is energetically coupled to mitochondrial TCA metabolism”.

### 2.7. Quantitative Real-Time Polymerase Chain Reaction (qRT-PCR) Analysis of Key Candidate Genes

To validate the DEGs identified from transcriptome analysis, six genes were randomly selected for a qRT-PCR assay. Primers were designed using the Primer3 software (v6.0; Appendix A). Following treatment at 4 °C, we isolated RNA from four samples of *A. truncatum* seeds and subsequently reverse transcribed it into cDNA templates. The reference gene Pact was utilized to verify the integrity of the template DNA, following which the experiment was initiated. To confirm the sequencing data’s reliability, we set a threshold for FPKM values exceeding 10 to pinpoint four genes that displayed significant variations, WRKY40(Atru.chr10.2411), WRKY53(Atru.chr9.2017), LEA14-A(Atru.chr2.3352), and CBF5(Atru.chr1.1259) (Figure 7A,B). We subsequently performed real-time quantification and concluded that the changes in the target gene at low temperature were consistent with the transcriptome data. CBF family genes are critical in the cold stress response. Typically, CBF family genes are induced under stress conditions, exhibiting differential expression patterns across tissues, predominantly under positive regulation. However, in this investigation, the CBF gene demonstrated negative regulation during seed germination, thereby complementing existing research on the CBF family (Figure 7C).

### 2.8. Network Relationships of Hub Genes in Four Modules with Significant Correlation

Transcription factors (TFs) play pivotal roles in various biological processes by modulating gene expression levels through binding to promoter regions. In our investigation, we identified 61 TF families involved in *A. truncatum* seed germination, with 14 TF families particularly crucial across all developmental stages. Notably, eleven of these families, such as LOB, GRAS, B3, bHLH, bZIP, EIL, MYB, MYB-related, NAC, TCP, and WRKY, are implicated in plant responses to biotic and abiotic stresses, including pathogens and environmental factors like low nitrogen, drought, salt, or cold stress [27,28,29,30,31]. Moreover, the core gene interaction network map elucidates the interactions among transcription factors. Notably, the pivotal role of WRKY family transcription factors during seed germination is evident, interacting with the LEA, ERF, and DER families (Figure 8). The WRKY family exhibits a more pronounced interaction with the ERF family. Both families contribute positively to biotic and abiotic stress responses in plants. Concurrently, the involvement of the LEA family facilitates the accumulation of late embryogenesis abundant proteins, providing essential resources for seed germination. These findings suggest that seed germination necessitates extensive metabolic activity, requiring the coordinated interaction of multiple transcription factor families to enhance seed viability and mitigate environmental stress, thereby establishing the foundation for subsequent seedling development.

## 3. Discussion

The response of plants to low-temperature stress encompasses both physiological and transcriptomic alterations. *A. truncatum*, an evergreen tree species, can grow normally at low-temperatures stress. In this study, we analyzed the physiological and transcriptomic modifications in *A. truncatum* seeds subjected to low-temperature stress. We aimed to understand its physiological and molecular mechanisms throughout growth and development, predict the key genes for these properties, and lay the foundation for the cultivation of other cold-tolerant plants.

To maintain cellular osmotic homeostasis, plants actively synthesize and accumulate proline as a compatible solute. This osmoprotective strategy increases intracellular solute concentration, thereby reducing water potential and facilitating enhanced water uptake capacity [32,33]. In this study, the proline and soluble protein contents of *A. truncatum* seeds under low-temperature conditions were consistently higher than those of seeds at room temperature (Figure 2A,B). This adaptive response enhanced the water retention capacity of the seeds (Figure 1C) and concomitantly reduced malondialdehyde (MDA) accumulation. Furthermore, plants employ sophisticated antioxidant defense systems to mitigate low-temperature-induced oxidative stress. These systems encompass key enzymes such as superoxide dismutase (SOD), peroxidase (POD), and catalase (CAT), which collectively scavenge reactive oxygen species (ROS), including hydrogen peroxide (H_2_O_2_) and superoxide radicals (O^2−^). By maintaining ROS at non-toxic levels, this enzymatic cascade effectively alleviates oxidative damage under cold stress conditions [34]. Consequently, the levels of antioxidant enzymes in plants exhibit a positive correlation with stress tolerance. However, the underlying regulatory mechanisms vary significantly across plant species. For instance, in tobacco (Nicotiana tabacum), enhanced cold tolerance is achieved through the coordinated upregulation of superoxide dismutase (SOD), ascorbate peroxidase (APX), and catalase (CAT) activities, which collectively mitigate oxidative damage during low-temperature stress [35]. In this study, *A. truncatum* seeds exhibited distinct temporal responses in antioxidant enzyme activities under low-temperature stress. Superoxide dismutase (SOD) activity showed significant elevation after 4 h of cold exposure, while peroxidase (POD) activity increased markedly following 24 h of treatment (Figure 3A,B). These initial enhancements in enzymatic activities suggest their crucial role in the cold stress response mechanism of *A. truncatum* seeds. However, prolonged stress exposure led to the progressive suppression of SOD activity. The glutathione (GSH) system demonstrated a time-dependent response, with GSH content increasing linearly throughout the treatment period (Figure 3C). This pattern reflects GSH’s continuous involvement in glutathione-mediated reduction reactions, which are essential for reactive oxygen species (ROS) scavenging in plant cells. The observed biphasic response pattern—characterized by initial activation followed by subsequent decline in SOD and POD activities—mirrors findings reported in cold-stressed apricot (Prunus armeniaca), suggesting a conserved antioxidant response strategy among woody plant species [36] and ginseng (Panax ginseng) [37]. Conversely, seedlings of Dali Tea (Camellia taliensis) exhibited a consistent decline in SOD activity [38], which is related to the physiological characteristics of the plant and may be a negative regulation in Dali Tea.

In the context of low-temperature stress, plant cells encounter an osmotic imbalance, precipitated by cellular dehydration. To counteract this stress, numerous plant species actively modulate endogenous osmoprotectants (e.g., proline, soluble sugars, and glycine betaine) to maintain cellular osmotic homeostasis and membrane stability [39]. In conditions of low temperature, soluble sugars (SS), soluble proteins (SP), and proline have been observed to accumulate as key osmoregulatory compounds. These compounds have been shown to play vital roles in the process of plant cold acclimation through multiple protective mechanisms [40]. In Camellia species, increased soluble sugar content shows a significant positive correlation with improved cold hardiness, suggesting its crucial role in cold adaptation mechanisms [41]. In *A. truncatum* seeds, prolonged cold exposure has been shown to result in a progressive accumulation of soluble protein, mirroring the cold adaptation response observed in *A. truncatum.* This phenomenon is further substantiated by studies on bilberry (Vaccinium vitis-idaea), where heightened soluble sugar levels exhibit a significant correlation with enhanced cold tolerance, thereby implying a conserved osmoregulatory strategy across diverse woody plant species [42]. The findings indicate that Acer rubrum may attenuate reactive oxygen species (ROS) levels through soluble sugar accumulation, thereby alleviating low-temperature-induced oxidative damage. This mechanism is analogous to observations in white clover (Trifolium repens), where elevated proline content has been experimentally demonstrated to enhance cold tolerance [43]. A similar response has been observed in oil palm (Elaeis guineensis), which exhibits a sustained increase in proline accumulation under chilling stress, consistent with the response pattern observed in *A. truncatum* seeds [44]. Notably, our investigation revealed a biphasic malondialdehyde (MDA) profile during prolonged cold exposure, characterized by initial accumulation followed by gradual decline (Figure 3D). This antagonistic relationship suggests that proline accumulation under abiotic stress may functionally counterbalance lipid peroxidation (as indicated by MDA levels), thereby maintaining cellular homeostasis and supporting physiological processes under stress conditions.

In response to abiotic stress, plants activate complex physiological and molecular adaptation mechanisms [45]. Transcriptomic profiling has emerged as a powerful approach for elucidating plant adaptive mechanisms under abiotic stress conditions [46,47,48]. Under conditions of low-temperature stress, analyses of GO and the KEGG in various plant species generally emphasize metabolic pathways, the synthesis of secondary metabolites, and the signaling of phytohormones [49,50]. Specifically, in *A. truncatum* seeds, the genes that exhibit the greatest differential expression are largely linked to metabolic pathways and the biosynthesis of secondary metabolites. KEGG showed that the differential genes during seed germination were mainly concentrated in carbon metabolism, TCA, circulation, the glycolytic cycle, and the synthesis of amino acids including pyruvate metabolism, alanine, aspartate, and glutamate metabolism, and glutathione metabolism (Figure 5). The TCA cycle functionally intersects with the glutathione cycle through three key nodes: (i) α-ketoglutarate provision for GSH synthesis, (ii) NADPH supply for GR activity, and (iii) citrate-derived acetyl-CoA for fatty acid elongation (Figure 6A,B). The enhancement of these life activities enhances seed vitality and promotes seed germination [51]. Such evidence implies that these essential regulatory pathways are vital for *A. truncatum* seeds’ adaptive response to low-temperature stress.

The regulatory role of transcription factors (TFs) is fundamental to plant cold adaptation mechanisms. Comparative genomic studies have identified evolutionarily conserved TF families that mediate low-temperature responses across phylogenetically diverse species, including bHLH55, AP2/EREBP56, MYB57, WRKY58, and NAC59. In *A. truncatum* seeds, the NAC gene family is represented more prominently than the ERF gene family. This investigation discovered a particular WRKY53 gene (Atru.chr9.2017) that could interact with essential genes (Figure 8). These findings set the foundation for subsequent functional validation studies of relevant genes.

## 4. Materials and Methods

### 4.1. Plant Materials and Low-Temperature Treatment

Two groups of *A. truncatum* seeds with equivalent weights (calculated based on 1000-seed weight) were surface-sterilized with a 3% sodium hypochlorite solution for 5 min. Thereafter, they were thoroughly rinsed with sterile distilled water 3–5 times. Subsequently, the seeds were aseptically placed on sterile filter paper in Petri dishes for subsequent experimental procedures. Two groups of *A. truncatum* seeds were placed in the light incubator with a light ratio of 16:8 and a temperature of 22 °C and 4 °C, respectively. Seed germination was observed and it was photographed when there were significant differences. The germination potential and the germination percentage were recorded.

### 4.2. Quantification of Seed Relative Water Content in A. truncatum

After removing the seed coats, *A. truncatum* seeds were allowed to germinate for three days. Subsequently, 10 seeds each from the low-temperature and room-temperature treatment groups were weighed (including seed and pericarp mass) using an analytical balance to record the fresh weight (FW). The seeds were then dried in an oven at 65 °C until a constant weight was achieved, which was recorded as the dry weight (DW).

The relative water content (RWC) of the seeds was calculated using the following formula:RWC = (1 − DWFW) × 100% RWC = (1 − FWDW) × 100%

### 4.3. Determination of Soluble Protein Content

The majority of soluble proteins in plants function as enzymes, which play essential roles in diverse metabolic processes. Quantifying their abundance provides a critical indicator for evaluating the overall metabolic activity of plant organisms. Transcriptome analysis of *A. truncatum* seeds under low-temperature stress revealed that most differentially expressed genes were enriched in carbon and amino acid metabolism pathways. Therefore, this study utilized Coomassie Brilliant Blue G-250 to determine protein content. This method relies on the chromogenic agent Coomassie Brilliant Blue G-250, which undergoes a spectral shift upon binding to hydrophobic regions of proteins. The resulting color intensity can be quantified by measuring the absorbance at 595 nm, with the optical density being directly proportional to the protein concentration [52,53].

### 4.4. Determination Method of Physiological Indicators

In this study, we conducted a rigorous analysis of the physiological responses exhibited by *A. truncatum* seeds, under low-temperature stress conditions of 4 °C and 25 °C. Several methods were used to accurately measure the levels of POD, SOD, GST, Pro, and MDA. SOD activity was measured by nitrogen blue tetrazolium; POD activity was measured by guaiacol; and MDA was measured by thiobarbital acid. Pro was measured using the acid ninhydrin chromogenic method: The sample extract was mixed with acid ninhydrin chromogen and color was developed in a boiling water bath. The absorbance was measured by a spectrophotometer. Proline reacts with a chromogen to generate a fluorescent material, whose fluorescence intensity is proportional to the proline content, which can be calculated by measuring the absorbance. Antioxidant enzymes play a crucial role in alleviating excess ROS. The activities of the antioxidant enzyme GST was measured using the test kit (Solarbio, Beijing, China) in accordance with protocols [54]. Three replicates of each index measurement were taken.

### 4.5. RNA-Seq Preparation and Assembly

Total RNA was extracted from both experimental groups using the TRNzol Universal RNA Kit (Tiangen Biotech, Beijing, China). RNA integrity was assessed through electrophoresis on 1% RNase-free agarose gels. To ensure the sequencing quality we used a Nano Photometer spectrophotometer to test the purity of RNA (OD 260/280 and OD 260/230 ratio); a Qubit2.0 Fluorometer to accurately quantify the RNA concentration; and an Agilent 2100 bioanalyzer to accurately detect RNA integrity.

After the enrichment of eukaryotic mRNA with polyA tails by magnetic beads with Oligo (dT), the mRNA was interrupted with a buffer. The first strand of cDNA was synthesized in the M-MuLV reverse transcriptase system using the fragmented mRNA as the template; the RNA strand was subsequently degraded with RNaseH and cDNA with dNTPs in the DNA polymerase I system. The purified double-stranded cDNA was subjected to end repair, A tail, and the ligation of sequencing joints, screened for about 200 bp of cDNA by AMPure XP beads, and PCR amplified, and AMPure XP beads were used to purify the PCR product again to obtain the final library. In order to ensure data quality, the original data should be filtered before the information analysis to reduce the analysis interference caused by the invalid data. The reads containing adapters were removed, and from the reads containing more than 10% N, low-quality reads with all A bases were removed (the number of bases of mass Q 20 was more than 50% of the whole read). After the data were filtered, we analyzed the composition and mass distribution of the bases to directly show the data quality to ensure that the base composition was relatively balanced, and then made the subsequent data monitoring more accurate. Bio informatic analysis was performed using Omicsmart, a dynamic real-time interactive online platform for data analysis (http://www.omicsmart.com). The transcriptome sequencing in this study was completed by Guangzhou Gidio Biotechnology Company on 25 February 2024.

### 4.6. Identification and Enrichment Analysis of Differentially Expressed Genes (DEGs)

The quantification of gene expression level was estimated by Fragments Per Kilobase of transcript per Million fragments mapped (FPKM). The differential expression analysis of each gene between pairwise comparisons of seed samples was performed using DESeq2 software (version 1.22.1) [55], according to fold change (FC) and false discovery rate (FDR), which were controlled by adjusting the *p* value through Benjamini and Hochberg’s approach. DEGs were identified using |log2FC| ≥ 1 and FDR < 0.01 as screening thresholds. GO functional enrichment and KEGG pathway enrichment analyses of DEGs were, respectively, implemented by ClusterProfiler packages based on Wallenius non-central hyper-geometric distribution [56] and KOBAS software (version 3.0) [57], with a significance level of *p* value < 0.05.

### 4.7. Validation of Key Candidate DEGs by qRT-PCR

To verify the reliability of transcriptomic data, 4 DEGs involved in seed germination responses to low temperatures were selected for qRT-PCR. The same RNA samples for transcriptome sequencing were reverse transcribed to cDNA using a HiScript III 1st Strand cDNA Synthesis Kit (+gDNA wiper) (R312-02, Vazyme, Nanjing, China). The qRT-PCR reaction system contained 10 µL of ChamQ SYBR Color qPCR Master Mix (Q411-2, Vazyme, Nanjing, China), 0.4 µL of forward primer, 0.4 µL of reverse primer, 2 µL of cDNA, and 7.2 µL of ddH2O, reaching a final volume of 20 µL. The qRT-PCR analysis was carried out on a CFX96 TOUCH instrument (Bio-rad, Hercules, CA, USA) with the following program: 95 °C for 30 s, followed by 40 cycles at 95 °C for 10 s, 60 °C for 30 s, and 72 °C for 30 s. All samples were examined in three biological replicates and three technological replicates. Pact was used as the reference gene. Relative expression levels of selected DEGs were calculated via the 2^−∆∆Ct^ method [58]. The primers were designed using Primer3 software (version 4.1.0) and the sequences are displayed in Appendix A.

### 4.8. Data Analysis

The experimental data were processed using GraphPad Prism (version 10.0), a software program designed for data analysis. The software performed multiple comparisons to identify significant differences. The software also generates additional graphs. A symbol “*” is used to indicate a significant result at the 0.05 level of confidence, and multiple “*” symbols indicate an extremely significant result at the 0.01 level.

## 5. Conclusions

In summary, the present study has made substantial contributions to the field of research concerning the molecular mechanisms underlying the response of A. truncatum seeds to low-temperature stress. Through a comprehensive analysis of the alterations in transcriptomic and physiological indicators under low-temperature stress conditions, it was observed that there were significant increases in MDA and Pro contents. Additionally, a distinctive pattern was identified, characterized by an initial increase followed by a subsequent decrease in the activities of SOD and POD enzymes. Additionally, the expression levels of related synthetic genes involved in the glutathione cycle and the TCA cycle were found to be significantly elevated. A substantial enrichment of KEGG pathways was observed in carbon metabolism, glycolysis, and the TCA cycle. These substances have been demonstrated to significantly enhance the process of seed germination, thereby rendering them a suitable subject for research related to low-temperature stress. These traits have been shown to enhance seed germination capacity, thereby establishing Acer truncatum as a model species for the study of low-temperature stress. Given the phytohormone-mediated regulation of germination, subsequent investigations require in vitro phytohormone induction assays. The findings demonstrate that proline application enhances seed germination under low-temperature conditions. This finding provides strategic insights for cultivating cold-tolerant agricultural practices through the optimization of germination processes.

## Figures and Tables

**Figure 1 ijms-26-11193-f001:**
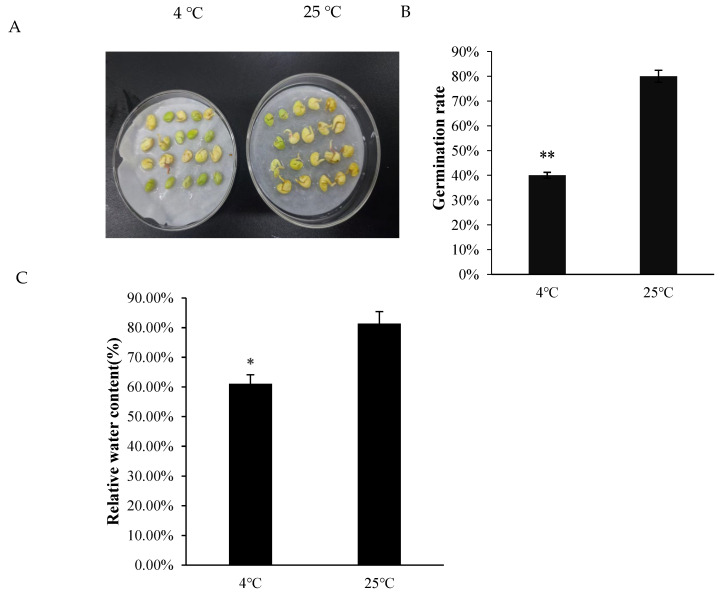
*A. truncatum* seed gemination and the varied penetrating substances. (**A**) Seed germination potential was compared at low temperature and normal temperature. (**B**) Seed germination rate of *A. truncatum.* (**C**) The changes in relative water content with low-temperature treatment. (* indicates significance, * indicates *p* < 0.05, and ** indicates *p* < 0.01).

**Figure 2 ijms-26-11193-f002:**
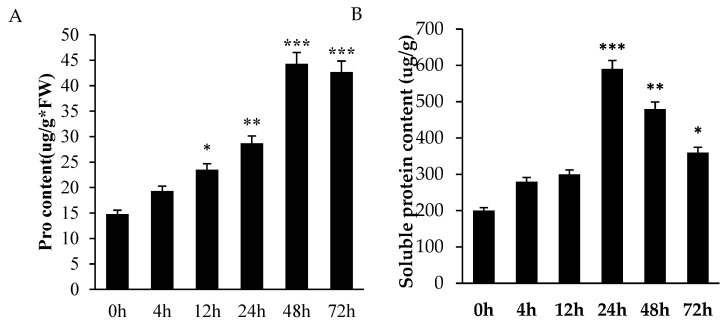
Significant shifts in soluble compound profiles were observed in seed cells during low-temperature stress. (**A**) The proline content in the seeds changed with cold treatment. (**B**) Soluble protein content with cold treatment. (* indicates significance, * indicates *p* < 0.05, ** indicates *p* < 0.01 and *** indicates *p* < 0.001).

**Figure 3 ijms-26-11193-f003:**
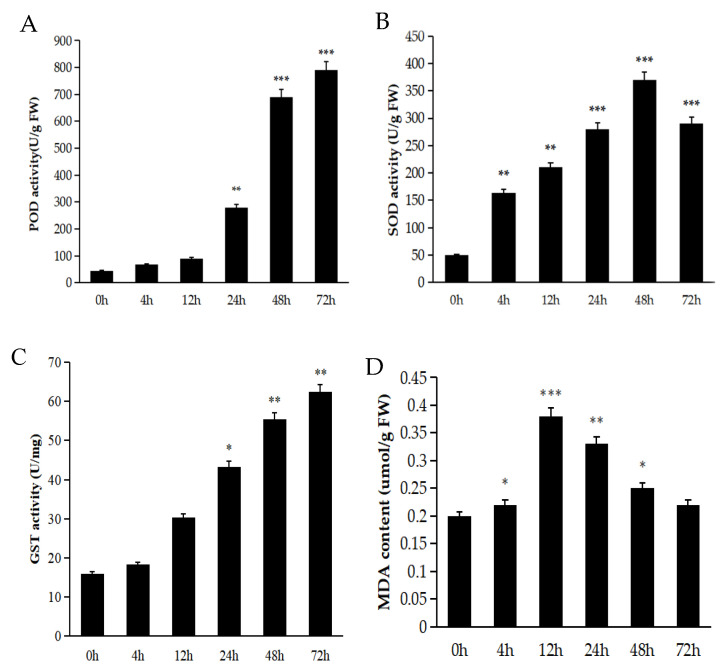
Changes in the antioxidant system of ingot maple seeds under low-temperature treatment (**A**) SOD (**B**) POD (**C**) GST (**D**) MDA. (* indicates significance, * indicates *p* < 0.05, ** indicates *p* < 0.01 and *** indicates *p* < 0.001).

**Figure 4 ijms-26-11193-f004:**
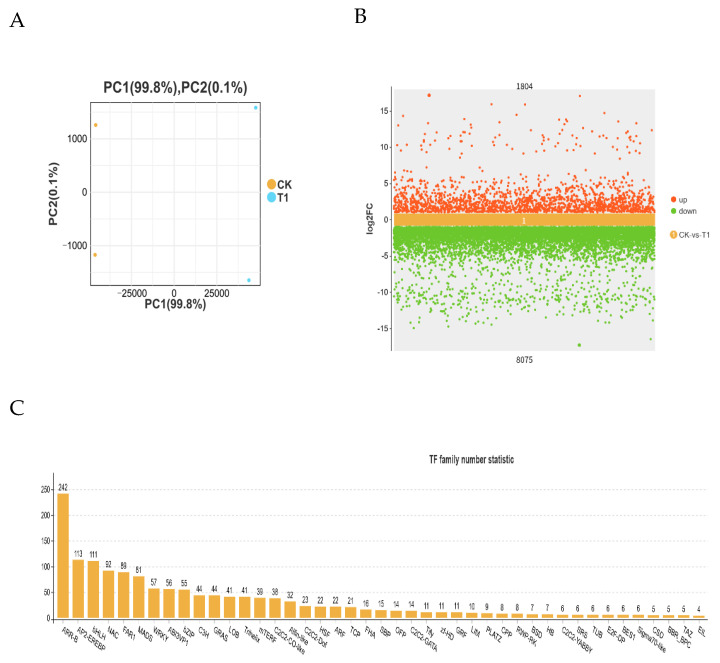
Transcriptome data for differential genes and differential transcription factors. (**A**) PCA analysis. (**B**) Differential genes between different groups. (**C**) Differential transcription factor statistics after low-temperature treatment.

**Figure 5 ijms-26-11193-f005:**
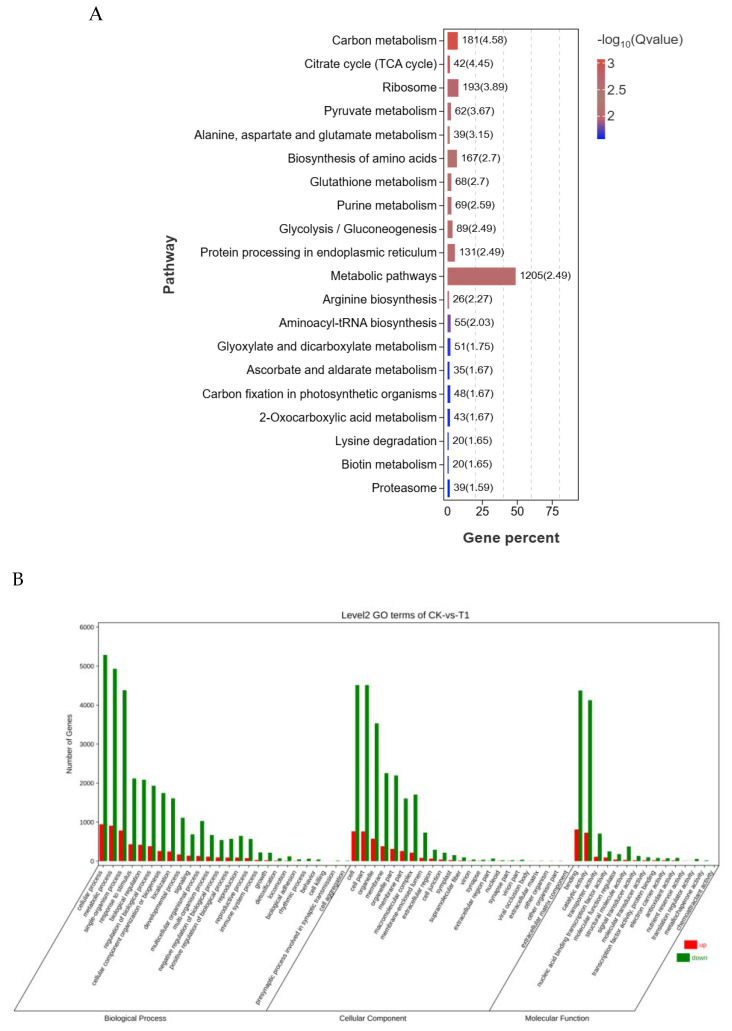
Functional enrichment of uniqueness. (**A**) KEGG pathway; (**B**) GO classification.

**Figure 6 ijms-26-11193-f006:**
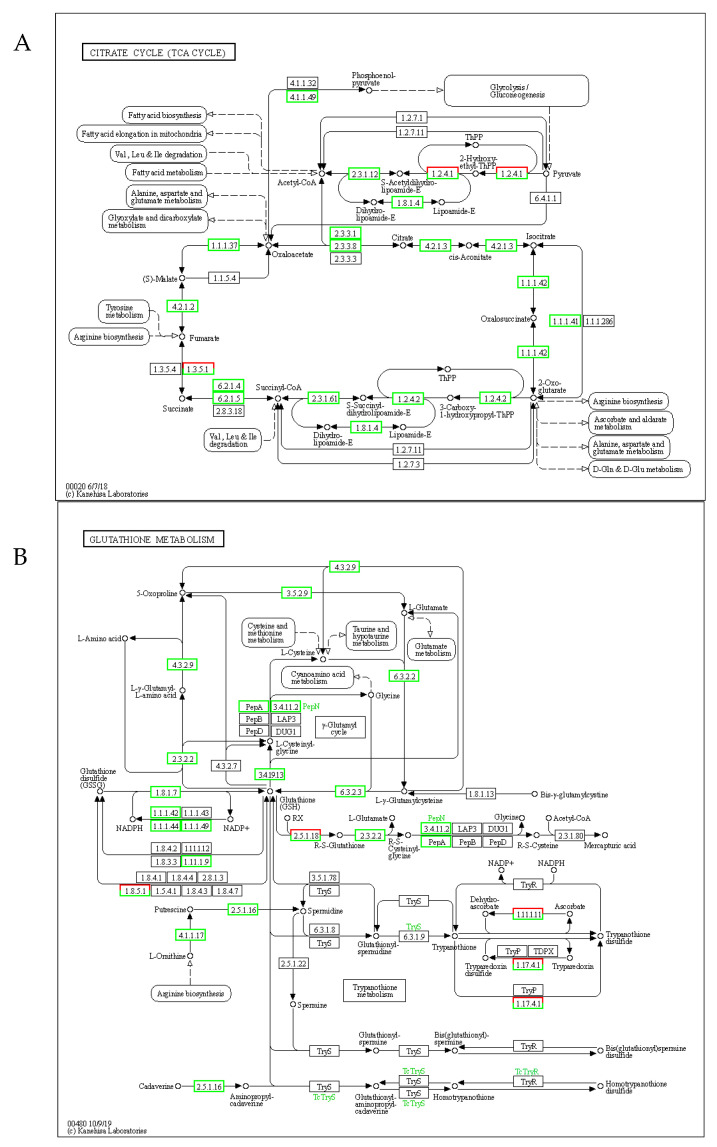
Integrated networks of the TCA cycle and glutathione redox metabolism in stress adaptation. (**A**) TCA cycle. (**B**) Networks of glutathione redox metabolism.

**Figure 7 ijms-26-11193-f007:**
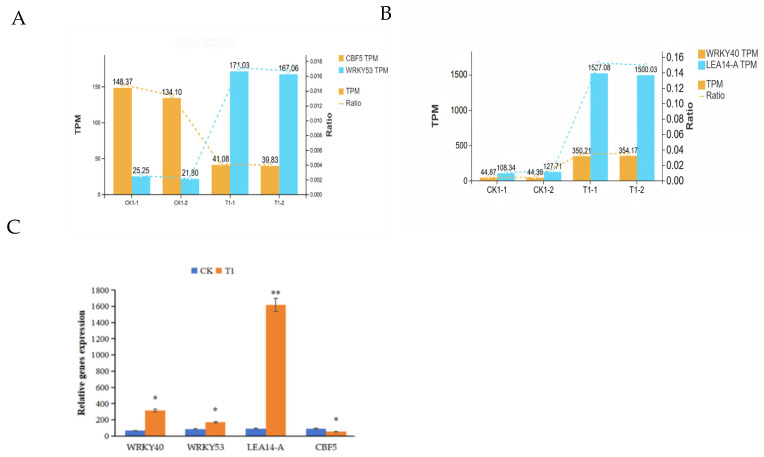
Expression levels and expression profiles of related hub genes. (**A**) The expression profile of CBF5 and WRKY53 between CK and T1. (**B**) The expression profile of WRKY40 and LEA14-A between CK and T1. (**C**) qRT-PCR of related hub genes. * indicates significance, * indicates *p* < 0.05, ** indicates *p* < 0.01).

**Figure 8 ijms-26-11193-f008:**
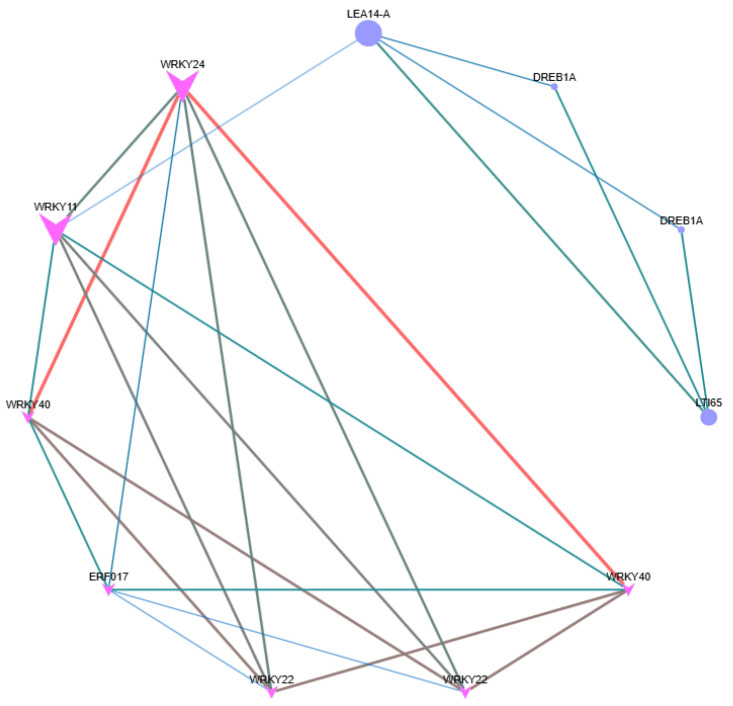
Network relationships of hub genes with significant correlation with different germination stages.

## Data Availability

Data supporting the findings of this work are available within this paper and its Appendix A files. The genetic materials used in this study are available from the corresponding authors upon request.

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
