# Peer review of "Transcriptome and Biochemical Analysis of the Mechanism of Low-Temperature Germination in Acer truncatum Bunge Seeds"

_ijms, 2025, doi:10.3390/ijms262211193_

Round 1

Reviewer 1 Report

Comments and Suggestions for Authors

This study provides information to better understand the physiological and molecular mechanisms underlying the response of Acer truncatum seeds to germinate under low temperature stress.

Although the manuscript provides numerous data, it contains several errors. Therefore, it cannot be accepted in its current form, and a major revision is necessary before acceptance for publication.

Below are some comments and suggestions that could improve the manuscript:

  1. In whole manuscript, Bunge should not be italic.
  2. Use the full name of Acer truncatum when mentioning for the first time, after that write A. truncatum.
  3. In abstract, L10: tolerance to what.
  4. Abstract contains long sentences that cause confusion and are difficult to understand. I suggest shortening these sentences to be easy to follow.
  5. L23-24: seed germination.
  6. L28: What does Pro stand for?
  7. L31: Replace “We guess” with “We assume” or “We propose”.
  8. The abstract is a little bit longer than normal.
  9. L46: Acer truncatum should be italic.
  10. L71: Rice (Oryzavsativa) and maize (Zea mays)
  11. L87: proline (Pro).
  12. L103: To this day,
  13. Throught the whole manuscript, all plant scientific names should be italic.
  14. L109-110: In this study,
  15. L116: seeds were
  16. L118-119: This sentence “The soluble protein content analysis” needs revision.
  17. L126-127: The lines are not bold.
  18. L122-124: The results of the soluble protein and proline contents should be moved under 2.2. not under 2.1.
  19. L127-131: This part that is related to references number 23, and 24 should be moved to the discussion section.
  20. L134: (Fig. 2A).
  21. L138, in the Figure 1 legends: delete “content”, and the means of significance (*p < 0.05, **p < 0.01*) needs revision.
  22. The soluble protein content was not mentioned in the text under subtitle 2.2. (Fig. 2B).
  23. L146: Give what ROS stand for.
  24. L147: What do you mean by “the transgenic plants”.
  25. L155-156: This sentence “under normal conditions“ needs revision.
  26. L156-159: This part that is related to reference number 25 should be moved to the discussion section.
  27. L169: Please verify what do you mean by “four samples”, especially in L368, you mentioned two groups.
  28. L174: Figure 3B needs revision.
  29. L226: 2.7. Quantitative real-time polymerase…..
  30. Fig. 7A, and Fig. 7B are not mentioned in the text.
  31. L246: 2.8. Network relationships….
  32. L366: How did you get the Acer truncatum seeds?
  33. L381: The formula needs revision.
  34. L382: Please explain in brief how soluble protein content was determined in the seeds.
  35. L405: Was this reference [54] used for the determination of only GST, or all measurements including POD, SOD, GST, Pro and MDA.
  36. L407: Please clarify what do you mean by 4 samples.
  37. The authors did not explain why the CBF genes demonstrated negative regulation during seed germination, while they are induced under stress conditions.
  38. In conclusion, it might be much better if you clarify the role of the results of this study in the agricultural sector, especially seed germination of crops.
  39. L490: Provide the accession number.

Comments on the Quality of English Language

1. There are some English typing errors.

2. The manuscript contains long sentences that should be shortened.

Reviewer 2 Report

Comments and Suggestions for Authors

Manuscript raises many interesting issues, however, I would like to ask you to respond to the comments.

  1. Double brackets when quoting,
  2. Inconsistency in the spelling of Acer truncatum Bunge - once in italics once not,
  3. Latin names used in the article should also be written in italics, 4.
  4. Please clarify the purpose of the study.
  5. ‘Plant cells typically enhance their water retention capacity by accumulating intracellular soluble compounds, thereby mitigating cellular damage under stress conditions’. - why is this written in bold?
  6. Not very technically elaborate manuscrytpt - e.g. figure 2.- why does the graph have a visible border, but other figures do not?
  7. Figure 2.- why is it written in italics? Please unify figure 2 A and B - the graphs are not made in the same way, it looks bad
  8. Figure 6 and 7 - not very readable, maybe figure can be enlarged?

9 Funding: Please add: ‘This research received no external funding‘ or “This research was funded by research and promotion of high value utilization technology ”, grant number (30802824)’ and The APC was funded by ‘ Monitoring of the Forest Quality Precision Improvement Project “grant number (30803179)”. Check carefully that the details given are accurate and use the standard spelling of funding agency names at https://search.crossref.org/funding. Any errors may affect your future funding. - Please sort this out, I think there are technical guidelines left on how to provide information about research funding.

  1. Refer to the literature notation requirements, please make corrections
  2. At which research centre were the analyses carried out? In what year? Years?
  3. How was germination carried out? How were the seeds laid out? On petri dish? - trays?

Round 2

Reviewer 1 Report

Comments and Suggestions for Authors

After carefully reviewing the modified manuscript and the corrections made by the authors, I believe that the authors have made the requested corrections, thus, the manuscript has improved considerably.

For this reason, the manuscript is now suitable for publication in IJMS in its current form.

Reviewer 2 Report

Comments and Suggestions for Authors

The authors have responded to the comments and corrected the suggested changes to the manuscript. The manuscript may be submitted for publication